# TASK2VEC READINESS: DIAGNOSTICS FOR FEDERATED LEARNING FROM PRE-TRAINING EMBEDDINGS

## ABSTRACT

Federated learning (FL) performance is highly sensitive to heterogeneity across clients, yet practitioners lack reliable methods to anticipate how a federation will behave before training. We propose readiness indices, derived from Task2Vec embeddings, that quantifies the alignment of a federation prior to training and correlates with its eventual performance. Our approach computes unsupervised metrics—such as cohesion, dispersion, and density—directly from client embeddings. We evaluate these indices across diverse datasets (CIFAR-10, FEMNIST, PathMNIST, BloodMNIST) and client counts (10–20), under Dirichlet heterogeneity levels spanning $\alpha \in \{0.05, \ldots, 5.0\}$ and FedAVG aggregation strategy. Correlation analyses show consistent and significant Pearson and Spearman coefficients between some of the Task2Vec-based readiness and final performance, with values often exceeding 0.9 across dataset×client configurations, validating this approach as a robust proxy for FL outcomes. These findings establish Task2Vec-based readiness as a principled, pre-training diagnostic for FL that may offer both predictive insight and actionable guidance for client selection in heterogeneous federations.

## 1 INTRODUCTION

Federated Learning (FL) (McMahan et al., 2017) has emerged as a central paradigm for collaborative model training under privacy and communication constraints (Zhang et al., 2021). However, its performance may be hindered by client heterogeneity, data imbalance, and federation size (Wen et al., 2023). Despite significant algorithmic advances in optimization and aggregation Yin et al. (2018); Pillutla et al. (2022); Xu et al. (2024), practitioners still lack principled tools to predict, before training, whether a given federation is likely to succeed. This absence of pre-training diagnostics forces costly trial-and-error experimentation and slows both research progress and deployment.

We introduce *Task2Vec Readiness*, a framework that leverages Task2Vec embeddings (Achille et al., 2019) to derive a quantitative readiness index for federated learning. Our approach transforms each client's data distribution into a fixed-dimensional embedding via Fisher Information, and then evaluates unsupervised metrics of federation structure. In particular, we measure cohesion (average cosine similarity among client embeddings), dispersion (average distance from the federation centroid), and density (RBF-kernel similarity over pairwise Euclidean distances). These metrics can be computed before training, and together they form a readiness profile that anticipates how well the federation can support collaborative optimization. The key novelty lies in repurposing task embeddings from transfer learning into a diagnostic signal for distributed training under heterogeneity.

We conduct extensive experiments across four benchmark datasets (CIFAR-10 (Krizhevsky et al., 2009), FEMNIST (Albaseer et al., 2021), PathMNIST (Yang et al., 2023), BloodMNIST (Yang et al., 2023)), with client counts ranging from 10 to 20 and Dirichlet non-IID partitions spanning $\alpha \in \{0.05, \ldots, 5.0\}$. Correlation analyses consistently reveal significant Pearson and Spearman coefficients between readiness metrics and final model performance, frequently exceeding 0.9 across dataset×client conditions. This validates readiness as a robust proxy for FL performance. Importantly, the results hold across different sources of heterogeneity, indicating that readiness captures structural properties of federations rather than dataset-specific artifacts.

Our contributions are twofold: (i) we introduce the first readiness index for federated learning based on task embeddings and (ii) we demonstrate its predictive validity across diverse datasets and heterogeneity regimes. By moving the focus from post-hoc evaluation to pre-training diagnostics, our framework not only provides a new lens for understanding federated learning dynamics, but also offers a practical tool for improving performance, guiding client selection, and enhancing the efficiency and reliability of federated training at scale.

## 2 BACKGROUND

### 2.1 FEDERATED LEARNING

Federated Learning (FL) is a machine learning paradigm in which a global model is trained collaboratively by multiple entities (which may represent individual devices or entire organizations) without sharing their private data (Aledhari et al., 2020). Unlike traditional centralized approaches, where all data is aggregated into a single infrastructure, FL enables training directly on decentralized datasets. This makes it especially valuable in highly regulated domains, such as healthcare and finance, where data protection laws restrict data sharing.

The prototypical FL setting consists of a central server $S$ and a set of distributed clients $C$, such that $|C| = K$, that jointly cooperate to solve a standard supervised learning task. Each client $c \in C$ has access to its own private training set $D_c = \{x_{c,i}, y_{c,i}\}_{i=1}^{n_c}$ where, $x_{c,i}$ denotes the $i$-th training sample of the $c$-th client, and $y_{c,i}$ denotes its corresponding label. The goal of FL is to train a global predictive model whose architecture and parameters $w \in \mathbb{R}_d$ are shared amongst all the clients and found to minimize $min_w \sum_{c=1}^{K} p_c L_c(w; D_c)$, where $L_c$ is the local objective and $p_c \geq 0$ specifies the individual contribution of the client $c$ such that $\sum_{c=1}^{K} p_c = 1$. Two possible configurations for $p_c$ are $p_c = \frac{1}{K}$ or $p_c = \frac{n_c}{N}$, where $N = \sum_{c=1}^{K} n_c$.

The local objective function $L_c$ usually is defined as the empirical risk calculated over the training set $D_c$ sampled from the client's local data distribution $L_c(w; D_c) = \frac{1}{n_c} \sum_{i=1}^{n_c} l(w; (x_{c,i}, y_{c,i}))$, where $l$ is an instance-level loss (e.g., cross-entropy loss or squared error in the case of classification or regression tasks, respectively).

In its standard form, Federated Learning employs Federated Averaging (FedAvg) (McMahan et al., 2017), which operates over iterative rounds:

1. The server $S$ selects a subset of clients $C^{(t)} \subseteq C$ and broadcasts the current global model $w^{(t)}$.

2. Each selected client updates $w^{(t)}$ locally on its data via stochastic gradient descent for multiple epochs, producing $\theta_c^{(t)}$.

3. The server aggregates updates using an aggregation function, typically a weighted average:

$$w^{(t+1)} = \sum_{c \in C^{(t)}} \frac{n_c}{N} w_c^{(t)}. \tag{1}$$

While FedAvg provides a simple and widely adopted baseline, its effectiveness in practice depends heavily on the properties of the participating clients. In particular, the global model's convergence and performance can be significantly affected by the statistical heterogeneity of client data, varying amounts of local data, and the total number of clients involved (Wen et al., 2023). These factors can lead to slower convergence rates or even instability during training, despite advances in robust aggregation methods (Yin et al., 2018; Pillutla et al., 2022; Xu et al., 2024).

### 2.2 TASK2VEC

Task2Vec is a method that was initially proposed "to provide vectorial representations of visual classification tasks which can be used to reason about the nature of those tasks and their relations" (Achille et al., 2019). In practice, in Task2Vec, a task is represented by a labeled dataset $D = \{(x_i, y_i)\}_{i=1}^{N}$. Then, the samples are passed through a pre-trained convolutional network, referred to as the probe network, and the representation of the task is captured by computing the diagonal of the Fisher Information Matrix (FIM) associated with its filter parameters.

Given a model with parameters $w$ and output distribution $p_w(y \mid x)$, the FIM is the expected covariance of the scores (gradients of the log-likelihood) with respect to the model parameters (Achille et al., 2019). It can be defined as:

$$F = \mathbb{E}_{x,y \sim \hat{p}(x,y)} \left[ \nabla_w \log p_w(y \mid x) \, \nabla_w \log p_w(y \mid x)^\top \right], \tag{2}$$

where $\hat{p}(x, y)$ denotes the empirical distribution of the dataset.

Since the full FIM may be prohibitively large depending on the probe network, Achille et al. (2019) introduce two approximations: (i) only the diagonal entries are retained, that is, it assumes that correlations between different filters in the probe network are not important, and (ii) the Fisher information is averaged across all weights within the same filter, since these weights are usually not independent. The resulting representation of the task therefore has fixed dimensionality, equal to the number of filters in the probe network. The extraction process is summarized below (adapted from (Achille et al., 2019)):

Given a labeled dataset $D = \{(x_i, y_i)\}_{i=1}^K$ and a fixed probe network $P$ (feature extractor), we compute the embedding as follows:

1. **Freeze $P$, train the classifier head.** Cache features $z = P(x)$ and fit a linear classifier $h_\theta$ on top, leaving the feature extractor weights fixed.

2. **Estimate Fisher on the feature extractor.** Excluding the classifier, compute a *diagonal* Fisher proxy for the parameters $w$ of $P$ using the *Monte Carlo (score matching):* iterate minibatches, forward $x$, sample labels $y \sim p_w(y \mid x)$, compute the loss $\ell(x, y)$, backprop, and *accumulate* squared gradients per weight (an estimator of the diagonal Fisher).

3. **Filter-level aggregation (fixed dimensionality).** For each convolutional filter, average the diagonal Fisher values across the weights belonging to that filter, yielding a single scalar per filter.

4. **Embedding.** Concatenate the per-filter scalars (classifier excluded) to obtain the task embedding $\mathbf{v} = \phi(D; P) \in \mathbb{R}^m$, where $m$ is the number of filters in $P$.

Broadly speaking, the FIM quantifies how sensitive the model's predictions are to small changes in its parameters. Directions in parameter space with high Fisher information correspond to features that are particularly important for solving the task, whereas directions with low values correspond to irrelevant or redundant features. By compressing this information into a fixed-length vector, Task2Vec creates an embedding that reflects both the difficulty of the task (through the norm of the embedding) and its similarity to other tasks (through distances between embeddings).

While originally designed to compare visual classification tasks in transfer learning, the same idea can be applied to federated learning by treating each client's local dataset as a distinct task. This enables us to compute Task2Vec embeddings at the level of federated clients, yielding a geometric portrait of the federation.

## 3 FEDERATION READINESS

We now introduce the notion of *federation readiness*, which provides a quantitative diagnostic of how well a federation is prepared to support collaborative optimization in federated learning.

**Definition.** Let $\mathcal{F} = \{D_1, D_2, \ldots, D_K\}$ denote a federation composed of $K$ clients, each holding a local dataset $D_c$. Given a fixed probe network $P$, we compute the Task2Vec embedding of each client's dataset, denoted by

$$\mathbf{v}_c = \phi(D_c; P) \in \mathbb{R}^m, \quad c = 1, \ldots, K, \tag{3}$$

where $\phi(D_c; P)$ is the Task2Vec embedding vector defined in Section 2.2.

The collection of client embeddings

$$V(\mathcal{F}; P) = \{\mathbf{v}_1, \mathbf{v}_2, \ldots, \mathbf{v}_K\} \tag{4}$$

forms the *federation embedding space*, which provides a geometric representation of the heterogeneity and structure of the federation.

**Federation Readiness Index.** A *readiness index* is a scalar functional

$$R : V(\mathcal{F}; P) \mapsto \mathbb{R}, \tag{5}$$

that maps the set of client embeddings to a real value measuring the expected ability of the federation to support collaborative learning.

The rationale is that a federation with high readiness corresponds to embeddings that are compact, balanced, and diverse enough to provide global coverage of the underlying task distribution. Such federations are structurally aligned and therefore more likely to achieve efficient collaborative optimization under standard aggregation rules (e.g., FedAvg). Conversely, low readiness signals that the federation suffers from excessive heterogeneity, imbalance, or sparsity, making convergence difficult and final performance unreliable. In the next section, we present candidates for the Federation Readiness Index.

## 3.1 READINESS INDICES

In this section, we present the tested readiness indices. First, we present the proposed *embedding-based indices*, derived directly from Task2Vec client embeddings. For comparison, we also present the federation entropy index, which captures statistical properties of the federation and has been used to characterize expected training difficulty Caldas et al. (2018); Solans et al. (2024).

### 3.1.1 EMBEDDING-BASED METRICS

**Cohesion.** Measures the average cosine similarity between all pairs of client embeddings:

$$\text{Cohesion} = \frac{1}{K(K-1)} \sum_{i \neq j} \cos(\mathbf{v}_i, \mathbf{v}_j). \tag{6}$$

Intuitively, high cohesion means that clients share similar feature relevance profiles, suggesting they can collaborate effectively with minimal conflict.

**Dispersion.** Captures the average distance of client embeddings to the federation centroid, normalized by the embedding dimensionality:

$$\text{Dispersion} = \frac{1}{K} \sum_{i=1}^{K} \frac{\|\mathbf{v}_i - \bar{\mathbf{v}}\|_2}{\sqrt{|\mathbf{v}_i|}}. \tag{7}$$

High dispersion indicates that clients are widely spread in embedding space, a signal of strong heterogeneity that may slow convergence. Since the goal is to quantify how well the federation is structured, from here on we use the negative of the dispersion as a readiness index.

**Density.** Quantifies the overall compactness of the federation via an RBF kernel applied to embedding distances:

$$\text{Density} = \frac{1}{K(K-1)} \sum_{i \neq j} \exp\left(-\frac{\|\mathbf{v}_i - \mathbf{v}_j\|_2^2}{2\sigma^2}\right), \tag{8}$$

where $\sigma$ is the median of pairwise distances. Federations with high density correspond to tightly packed embeddings, suggesting stable aggregation dynamics.

**Cohesion–Dispersion Index (CDI).** We define the Cohesion–Dispersion Index (CDI) as a weighted combination of cohesion and dispersion:

$$\text{CDI}_{\beta,\gamma} = \beta \cdot \text{Cohesion} + \gamma \cdot (-\text{Dispersion}), \tag{9}$$

where $\beta, \gamma > 0$ are weighting coefficients.

This index balances the tendency of clients to align with one another (high cohesion) against the degree to which they are scattered in embedding space (low dispersion). A high CDI value therefore indicates a federation that is both well-aligned and structurally compact, properties expected to facilitate stable and efficient collaborative training. In this work, we set $\beta = 1$ and $\gamma = 1000$ to scale the dispersion term to the same order of magnitude as cohesion.

### 3.1.2 DISTRIBUTIONAL INDEX

**Average Entropy.** Measures the average label entropy across clients:

$$\text{AvgEntropy} = \frac{1}{n} \sum_{i=1}^{n} H(p_i), \quad H(p_i) = - \sum_{c \in \mathcal{C}_i} p_{i,c} \log p_{i,c}. \tag{10}$$

High entropy means that client data are locally balanced across classes, whereas low entropy reflects strong label imbalance.

## 4 EXPERIMENTAL SETUP

We designed an experimental pipeline to evaluate the relationship between Task2Vec-based readiness and federated learning (FL) performance under different degrees of data heterogeneity. The experiments are implemented in Python and organized to sweep datasets, federation sizes, and Dirichlet heterogeneity parameters.

**Datasets** We evaluate our approach using four benchmark datasets covering both natural image classification and medical image analysis tasks:

- **CIFAR-10** (Krizhevsky et al., 2009): An image dataset with 60,000 RGB images of size $32 \times 32$ pixels, equally distributed across 10 classes.
- **FEMNIST** (Albaseer et al., 2021): A federated extension of the EMNIST dataset consisting of grayscale handwritten characters, naturally partitioned by writer.
- **PathMNIST** and **BloodMNIST** (Yang et al., 2023): Two medical imaging datasets from the MedMNIST collection. PathMNIST 100,000 histopathology images of colorectal tissue, spanning 9 classes, while BloodMNIST 17,092 microscopic images of blood cells, spanning 8 classes.

For CIFAR-10 and MedMNIST, synthetic non-IID (not independent and identically distributed (IID)) scenarios are generated by partitioning data across clients using a Dirichlet distribution over class labels, controlled by the concentration parameter $\alpha$. A low $\alpha$ means that clients have data from only a few classes (highly non-IID). Meanwhile, a high $\alpha$ implies that the data distribution approaches IID. For FEMNIST, we maintain the natural writer-based partitioning which is naturally non-IID.

**Preprocessing** All images are normalized according to their dataset-specific mean and standard deviation. For Task2Vec analysis, all samples are converted to 3-channel RGB and resized to 224×224, ensuring compatibility with the probe network.

**Baseline Model** In all experiments, federations use a fixed ResNet34 architecture (He et al., 2016; PyTorch, 2025) with randomly initialized weights. This residual network is chosen for its strong performance in image classification, reasonable size, and its ability to generalize across domains.

Modifications are made only to: (i) Initial convolutional layer: adapted for single-channel datasets (e.g., FEMNIST), and (ii) final fully connected layer: adjusted to match the number of classes in each dataset. By using the same core architecture, we ensure that variations in performance are attributable to data distributions and federation parameters rather than architectural differences.

**Federation Design** To systematically evaluate readiness under controlled conditions, we designed federations that vary along two main axes: the number of clients and the degree of statistical heterogeneity. The setup is summarized below.

- **Number of Clients** ($N$): $\{10, 20\}$.
- **Optimizer:** SGD (momentum = 0.9).
- **Learning rate:** 0.01.

- **Communication Rounds** ($T$): 20 communication rounds.
- **Aggregation Strategy:** Federated Averaging (FedAvg) is the the global aggregator. In each round:
  - All clients participate in local training (`fraction_fit`=1.0).
  - A random half of the clients (`fraction_evaluate`=0.5) are sampled for evaluation.
- **Local Training:** Each client executes $E = 1$ local epochs per round using a batch size of $B = 32$.
- **Heterogeneity Simulation:** For the CIFAR-10 and MedMNIST datasets, non-IID splits are generated by drawing per-client class proportions from a Dirichlet distribution with concentration parameter

$$\alpha \in \{0.05,\, 0.1,\, 0.2,\, 0.3,\, 0.5,\, 1.0,\, 2.0,\, 5.0\}\,.$$

**Evaluation Metrics**   Model performance is assessed using Top-1 accuracy for CIFAR-10 and FEMNIST, and macro-averaged AUC for PathMNIST and BloodMNIST to account for class imbalance. Final results are reported on a centralized test set reserved exclusively for evaluation after training.

**Task2Vec Readiness Computation**   Before starting federated training, Task2Vec embeddings are computed for each client's dataset using the ResNet34 architecture as a probe network, $P$, pretrained on IMAGENET1K_V1. The embeddings are then combined to compute the Readiness Index.

Task2Vec embeddings are computed using the following parameters:

- Maximum samples for embedding computation: 1,000 per client.
- Skip layers: 6 (excluding early feature layers).
- Embedding dimension: Fixed at 1,000 features.
- Probe network: ResNet34.

**Implementation Details**   The entire experimental pipeline is implemented in Python 3.9. The following key libraries and frameworks have been used:

- **PyTorch 2.5.1+cu121** for model definition, local training, and evaluation.
- **Flower (flwr) 1.20.0** for federated orchestration, including client-server communication and FedAvg aggregation.
- **NumPy 2.1.2** and **SciPy 1.16.1** for data manipulation and statistical computations.
- **scikit-learn 1.7.2** for additional metrics and analysis.
- **Task2Vec set (by AWS)**[1] for pre-training task embeddings.

**Hardware Setup**   Experiments were executed on a single machine equipped with:

- 1 GPU (NVIDIA RTX-class).
- 20 CPU cores allocated to clients.
- 128 GB of RAM.

## 5   RESULTS AND DISCUSSION

Table 1 presents Pearson correlations between final performance at the different heterogeneity levels and the different readiness metrics. Correlations and $p$-values were computed using the `pearsonr` function from the `statsmodels` library, which performs a two-sided test of the null hypothesis that the true correlation is zero. Following standard practice, we consider correlations statistically significant at the $\alpha = 0.05$ level (bolded in the table).

---

[1]Available at `https://github.com/awslabs/aws-cv-task2vec` Accessed: 2025-09-23

The results show that cohesion and −dispersion (i.e., the inverted dispersion score) are both highly predictive across nearly all Dataset×$K$ settings, with correlations frequently above 0.95 and significant $p$-values. The Cohesion–Dispersion Index (CDI) achieves similarly strong values, confirming that the combined signal of alignment and compactness is a stable indicator of training outcomes. By contrast, density remains inconsistent: it is weakly positive for some cases (PathMNIST with $K = 10$) but negative or non-significant elsewhere (e.g., PathMNIST with $K = 20$). The federation entropy baseline also correlates well with performance, often in the 0.80–0.95 range, but tends to underperform the embedding-based metrics. Overall, the table highlights that embedding-derived measures, particularly cohesion, −dispersion, and CDI, are the most reliable predictors of final federated accuracy.

Table 1: Pearson correlation between final performance and readiness metrics by Dataset×K

| Dataset | K | Cohesion | -Dispersion | Density | CDI | Average Entropy |
|---------|---|----------|-------------|---------|-----|-----------------|
| bloodmnist | 10 | **0.994(0.000)** | **0.794(0.019)** | -0.605(0.112) | **0.940(0.001)** | **0.825(0.012)** |
| bloodmnist | 20 | **0.958(0.000)** | **0.868(0.005)** | 0.227(0.589) | **0.940(0.001)** | **0.881(0.004)** |
| cifar10 | 10 | **0.964(0.000)** | **0.966(0.000)** | 0.157(0.710) | **0.982(0.000)** | **0.950(0.000)** |
| cifar10 | 20 | **0.845(0.008)** | **0.985(0.000)** | -0.217(0.606) | **0.952(0.000)** | **0.981(0.000)** |
| femnist | 10 | **0.953(0.000)** | **0.795(0.018)** | -0.152(0.720) | **0.852(0.007)** | **0.857(0.007)** |
| femnist | 20 | **0.989(0.000)** | **0.988(0.000)** | -0.386(0.345) | **0.991(0.000)** | **0.899(0.002)** |
| pathmnist | 10 | **0.909(0.002)** | 0.646(0.083) | 0.599(0.116) | **0.776(0.023)** | **0.727(0.041)** |
| pathmnist | 20 | **0.931(0.001)** | **0.861(0.006)** | **-0.877(0.004)** | **0.917(0.001)** | **0.876(0.004)** |

Cells show $r(p)$; bold indicates $p < 0.05$.

Table 2 reports the Spearman rank correlations between final performance at the different heterogeneity levels and the readiness metrics. Correlations and $p$-values were computed using the `spearmanr` function from the `statsmodels` library, which evaluates the monotonic association between two variables and also performs a two-sided test of the null hypothesis that the correlation is zero. As before, we adopt $\alpha = 0.05$ as the significance threshold (bolded in the table).

The results largely confirm the Pearson analysis: cohesion and −dispersion are consistently strong predictors across all datasets and federation sizes, with correlations above 0.90 and highly significant $p$-values. The Cohesion–Dispersion Index (CDI) remains stable as well, showing values that track closely with the individual metrics. Interestingly, federation entropy also yields high correlations, often matching or slightly exceeding the embedding-based measures in smaller federations (e.g., FEMNIST with $K = 10$). Density, again, shows inconsistent behavior: in some cases it is weakly negative or non-significant, while in PathMNIST with $K = 10$, it unexpectedly emerges as positively correlated ($r = 0.905$, $p = 0.002$). Taken together, the Spearman results strengthen the conclusion that embedding-based readiness indices—particularly cohesion, −dispersion, and CDI—are robust predictors of FL outcomes, capturing not just linear but also monotonic relationships.

Table 2: Spearman correlation between final performance and readiness metrics by Dataset×K

| Dataset | K | Cohesion | -Dispersion | Density | CDI | Average Entropy |
|---------|---|----------|-------------|---------|-----|-----------------|
| bloodmnist | 10 | **0.976(0.000)** | **0.929(0.001)** | -0.119(0.779) | **0.976(0.000)** | **0.976(0.000)** |
| bloodmnist | 20 | **0.976(0.000)** | **0.976(0.000)** | -0.238(0.570) | **0.976(0.000)** | **0.976(0.000)** |
| cifar10 | 10 | **0.952(0.000)** | **0.976(0.000)** | -0.095(0.823) | **0.976(0.000)** | **0.952(0.000)** |
| cifar10 | 20 | **0.970(0.000)** | **0.970(0.000)** | -0.323(0.435) | **0.970(0.000)** | **0.970(0.000)** |
| femnist | 10 | **0.905(0.002)** | **0.786(0.021)** | -0.119(0.779) | **0.881(0.004)** | **0.952(0.000)** |
| femnist | 20 | **1.000(0.000)** | **0.905(0.002)** | -0.429(0.289) | **0.976(0.000)** | **1.000(0.000)** |
| pathmnist | 10 | **0.786(0.021)** | **0.738(0.037)** | **0.905(0.002)** | **0.786(0.021)** | **0.786(0.021)** |
| pathmnist | 20 | **0.881(0.004)** | **0.857(0.007)** | -0.690(0.058) | **0.881(0.004)** | **0.881(0.004)** |

Cells show $r(p)$; bold indicates $p < 0.05$.

Interestingly, the Task2Vec-based indices (cohesion, −dispersion, and CDI) showed stronger correlations with final performance under the Pearson measure compared to Spearman. This indicates that embedding-derived metrics preserve a more *linear* relationship with federated accuracy: as clients

become more aligned in embedding space, accuracy tends to increase in a nearly proportional fashion. In contrast, purely distributional measures such as federation entropy may capture only broader monotonic trends (e.g., more balanced class distributions usually help).

This suggests that the Fisher-information geometry underlying Task2Vec is not just a diagnostic signal of heterogeneity, but also aligns closely with the linear mechanisms through which heterogeneity affects convergence speed and final accuracy. Practically, this implies that embedding-based readiness indices could serve not only as robust predictors, but also as quantitative levers. That is, small changes in cohesion or dispersion may map linearly to predictable gains or losses in model performance, enabling more precise client selection or weighting strategies in real federations.

One, perhaps, tangential finding is that density, as currently defined, did not perform as well as the other embedding-based metrics. At first glance, this is surprising since a dense federation sounds like it should be a good one. But density, as we compute it, is driven by local compactness—how tightly small groups of clients cluster together—rather than the global arrangement of the whole federation. A federation with several tight but far-apart clusters can end up with high density, even though those clusters are poorly aligned and FedAvg will struggle. Cohesion and dispersion, by contrast, reflect the overall geometry. They tell us how similar clients are on average and how widely they spread around the centroid. This global perspective aligns much better with the dynamics of aggregation, which explains why cohesion and dispersion (and especially their combination in CDI) give more stable and predictive signals of readiness. That said, density should not be dismissed: future work could revisit it with adaptive kernel scales or multi-scale definitions, which may better capture the interplay between local clustering and global alignment in federations.

## 6 CONCLUSION

We introduced *Task2Vec Readiness*, a framework for diagnosing the likelihood of success in federated learning (FL) before training. By embedding each client's dataset into a Fisher-information geometry and defining indices such as cohesion, $-$dispersion, and the Cohesion–Dispersion Index (CDI), we showed that embedding-based measures correlate strongly with final performance across four datasets and varying heterogeneity levels, with Pearson and Spearman coefficients often above 0.9.

Compared to distributional statistics such as average entropy, Task2Vec-based indices generally offered stronger and more stable signals. Entropy remains useful for capturing label balance, but embeddings reflect feature-level geometry, aligning more closely with optimization dynamics and yielding more linear, predictive relationships with accuracy.

The main limitations of this study are: we fixed a single probe network (ResNet34), focused on small-to-moderate federations in image classification, and evaluated only under FedAvg. Future work should test alternative probes, larger and more diverse federations, other aggregation strategies, and refined density definitions that better capture both local clustering and global alignment. Embedding-based diagnostics remain a promising path toward principled, pre-training readiness tools for federated learning.

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

## A  APPENDIX

### USE OF LARGE LANGUAGE MODELS (LLMs)

Large Language Models (LLMs) were used as assistive tools in the preparation of this paper. Specifically, LLMs (ChatGPT, GPT-5) supported (i) refining the writing style for clarity and flow, (ii) suggesting alternative phrasings and section transitions, and (iii) drafting preliminary descriptions of methods, metrics, and discussion points based on author-provided inputs and results. All research ideas, experimental design, data analysis, and scientific claims originated from the authors. The authors take full responsibility for the final content of the paper.

