# OpenReview forum: "Task2Vec Readiness: Diagnostics for Federated Learning Performance from Pre-Training Embeddings"
_ICLR.cc/2026/Conference — Submitted to ICLR 2026_

### Official Review · Reviewer_LmY5 · 2025-10-21

**Soundness:** 2
**Presentation:** 2
**Contribution:** 2
**Rating:** 2
**Confidence:** 3

**Summary:**

- The authors use a probe network to get the embedding vectors from all clients' data.
- From these embedding vectors, the authors calculate Cohesion (cosine sim. between clients), Dispersion (distance from centroid), and Density (kernel similarity between embeddings)
- The authors claim these metrics are correlated with final performance.

**Strengths:**

- Being able to predict the performance of Federation would be extremely useful. If the metrics are indeed accurate, it would provide useful insight, to judge whether FL is a viable option with the data that is present.
- The authors do well in selecting various datasets (4) while also varying the number of clients and the heterogenicity.

**Weaknesses:**

- My main concern with this paper is the transparency of the results. The only results are Table 1 and 2. However, they only provide the correlation (spearman vs pearson). They do not provide the actual accuracies or the actual metrics that are derived from their experiments. This makes the actual review of the paper quite difficult. I'm not sure what I'm supposed to review if I cannot even see the main results. The authors also do not provide any intuition or visualization of the metrics that have been chosen.
- For example, a high correlation might be statistically significant but practically meaningless if the performance variation across all experiments is very small (e.g., all accuracies are between 91% and 92%).
- Minor points: the paper does not contain any figures, which could have been used to help the reader understand the method better. Additionally, the paper is less than 8 pages, which is far below the 9 page-limit. The authors should use the space more wisely.

**Questions:**

- Can the authors provide the actual results derived from their experiments?
- Can the authors provide the intuition and visualization of how and why the three metrics, Cohesion, Dispersion, and Density were selected?

---

### Official Review · Reviewer_B2u5 · 2025-10-30

**Soundness:** 2
**Presentation:** 1
**Contribution:** 2
**Rating:** 2
**Confidence:** 4

**Summary:**

The paper treats the data distribution of each client in federated learning as an independent "task" and uses the Task2Vec method to generate a fixed-dimensional embedding vector for each client. These embedding vectors together constitute the geometric representation of the entire federation.

**Strengths:**

The paper cleverly reapplies the Task2Vec embedding technique, which originates from transfer learning, to the diagnostic problem of federated learning, which is a novel and insightful perspective.

**Weaknesses:**

The paper suffers from several shortcomings:

**Construction is simplistic:** The paper reads more like an experimental report, failing to clearly describe the experimental motivation and methodology.

**Single Probe Network Selection:** The entire study relies on a single pre-trained ResNet34 model as the probe network for Task2Vec. The paper does not explore whether using different architectures (such as ViT) or pre-training the probe network on different datasets affects the effectiveness and generalization ability of the readiness index, which could be a potential limitation of the method.

**Small Federated Scale:** The experiments only cover scenarios with 10 and 20 clients. This is sufficient for validating the concept, but real-world federated learning systems may involve hundreds, thousands, or even millions of clients. The scalability and predictive performance of this method under extremely large federated scenarios remain to be verified.

**Single Aggregation Strategy:** The study uses only the standard FedAvg algorithm for validation. FedAvg is known to face challenges with highly non-IID data. The paper does not examine whether the readiness index is equally applicable to predicting the performance of other more advanced aggregation algorithms designed specifically for handling heterogeneity.

**Questions:**

See Weaknesses

---

### Official Review · Reviewer_2Th8 · 2025-10-30

**Soundness:** 1
**Presentation:** 2
**Contribution:** 2
**Rating:** 2
**Confidence:** 3

**Summary:**

This paper introduces Task2Vec Readiness, a framework designed to predict the performance of a federated learning (FL) system before training. The method leverages Task2Vec embeddings (Achille et al., 2019) to represent each client’s dataset as a vector in Fisher-information space, from which several unsupervised metrics such as cohesion, dispersion, density, and a combined Cohesion–Dispersion Index (CDI), are computed. These metrics are intended to serve as readiness indicators for a federation, quantifying how “aligned” the clients are and thus how likely the federation is to train effectively.

Experiments on four datasets (CIFAR-10, FEMNIST, PathMNIST, BloodMNIST) and varying non-IID levels show strong correlations between readiness metrics and final FL performance. The authors claim that these measures offer actionable pre-training diagnostics that could guide client selection or federation design.

**Strengths:**

+ This paper shows some experimental results with multiple datasets and Dirichlet non-IID settings.
+ The paper identifies an interesting diagnostic direction in FL literature.

**Weaknesses:**

1. The novelty and technical depth is limited. The method is primarily an application of Task2Vec to FL, without new theoretical or algorithmic contributions. The readiness indices (cosine similarity, Euclidean distance, kernel density) are straightforward and commonly used; no new diagnostic measure is introduced beyond simple statistics on embeddings. The connection between these metrics and actual FL convergence is intuitive but not theoretically supported.

2. This paper lacks theoretical foundation or analysis. There is no formal analysis demonstrating that the proposed indices can predict convergence or stability, only empirical correlations. The paper claims “readiness” reflects structural properties of the federation, but provides no analytical link between Fisher geometry and optimization dynamics. Without a theoretical underpinning, the method risks being perceived as a heuristic correlation study rather than a principled diagnostic framework.

3. The only comparison is with entropy-based statistics. There are no baselines from prior works that predict FL performance or data heterogeneity metrics (e.g., Dirichlet α, earth mover distance, gradient divergence metrics). Without such comparisons, it is hard to tell whether Task2Vec embeddings offer tangible advantages over simpler or computationally cheaper heterogeneity measures.

4. The experiments are insufficient. It is unclear whether the correlations generalize across probe networks or FL algorithms (only FedAvg is tested). The small federation sizes (10–20 clients) and fixed architecture limit generalizability.

**Questions:**

1. The proposed framework applies Task2Vec to federated learning without introducing new metrics beyond standard similarity and distance measures. Could you elaborate on what new insights or theoretical contributions distinguish your approach from simply computing existing heterogeneity statistics (e.g., pairwise cosine similarity of client features)?

2. Can you provide a theoretical or analytical justification linking your readiness indices (cohesion, dispersion, density) to FL convergence behavior or optimization stability? For example, is there any mathematical relation between these metrics and gradient divergence or convergence bounds?

3. Could the authors add more literature comparison?

---

### Official Review · Reviewer_hxTt · 2025-10-31

**Soundness:** 1
**Presentation:** 1
**Contribution:** 2
**Rating:** 2
**Confidence:** 4

**Summary:**

This paper proposes “Task2Vec Readiness,” a suite of diagnostic indices derived from Task2Vec embeddings, to estimate the "readiness" of federated learning (FL) federations before training. By embedding each client’s local data distribution into a fixed-length representation (using Fisher Information from a probe network), the authors extract several geometric indices (cohesion, dispersion, density, and composite CDI) intended to capture how well a federation is structurally prepared for successful collaborative learning.

**Strengths:**

The paper focus on an interesting problem in FL to predict federation success before incurring costly FL runs—by leveraging task embeddings.

**Weaknesses:**

1. The overall quality of this paper should be improved, e.g., the structure and writing of this paper. This paper also lacks the necessary explanation about the notions, e.g., of the meaning of $\phi$
2.  All experiments are conducted with up to 20 clients and mostly using standard benchmarks partitioned synthetically (via Dirichlet splits). This raises doubts about the external validity and practical robustness of the readiness indices.
3. The entire experimental pipeline is limited to the FedAvg algorithm and ResNet34 as the probe network. While this controls for variables, it makes it unclear whether the proposed indices generalize to other aggregation schemes, model architectures, or tasks beyond image/classification. There is no ablation or evidence for robustness to the choice of probe network or optimizer.
4. While Tables 1 and 2 report impressive correlations (Spearman/Pearson >0.9 in many settings) between readiness indices and final performance, the deployment utility is less clear. There is no demonstration that these metrics can meaningfully inform *actionable* pipeline decisions (e.g., client selection, federation pruning, or adjusting hyperparameters to boost readiness).
5. It is suggested that the author also report the experimental results to provide a more intuitive perception of the direct relationship between the indicators and performance.
6. All readiness indices are computed over Task2Vec embeddings with 1,000 examples per client, fixed dimension of 1,000, and a ResNet34 probe. There is no examination of how sample size, probe architecture, or dimensionality affect the robustness or informativeness of the readiness indices.
7. While Table 1 and Table 2 contain exact correlation coefficients and associated p-values, no confidence intervals, scatter plots, regression residuals, or error analyses are provided.

**Questions:**

See above

---

### Meta-Review · Area_Chair_gvzq · 2025-12-29

**Summary:**

The paper introduces a framework called Task2Vec Readiness, designed to predict the performance of federated learning (FL) systems prior to training. By leveraging Task2Vec embeddings derived from individual clients' datasets using Fisher Information geometry, the authors compute several geometric indices: cohesion, dispersion, and density. These indices form a readiness profile that serves as a diagnostic tool for assessing the likelihood of successful federated training. The authors validate their approach through experiments on multiple datasets, showing strong correlations between readiness indices and final model performance.

Reviewers' Concerns:

Lack of Novelty and Theoretical Insights: Reviewers noted that the approach mainly applies existing Task2Vec techniques to FL without introducing new theoretical contributions. Critics argued that the diagnostic measures are basic and commonly used, lacking innovative insights.

Experimental Limitations: The experiments were primarily conducted with a limited number of clients (10-20) and only with one aggregation algorithm (FedAvg). Reviewers expressed concerns about the generalizability of the findings to larger federations and different contexts, such as varying models or optimization strategies.

Insufficient Analysis and Reporting: Reviewers highlighted the absence of detailed performance metrics, visualizations, and a transparent reporting of results beyond correlation coefficients. The lack of comparisons to other heterogeneity metrics or methods was also concerning, as it made it difficult to assess the advantages of the proposed readiness indices.

Ambiguous Practical Utility: While the authors claimed that the readiness indices could inform actionable decisions (like client selection), reviewers found no evidence in the paper to demonstrate how these metrics would guide meaningful changes in practice.

**Reviewer Concerns:**

There was no rebuttal by the authors.

**Reviewer Scores:**

There was no rebuttal by the authors, and thus the authors didn't engage with the reviewers.

---

### Decision · Program_Chairs · 2026-01-26

Reject